# Shoot Regeneration Is Not a Single Cell Event

**DOI:** 10.3390/plants10010058

**Published:** 2020-12-29

**Authors:** Patharajan Subban, Yaarit Kutsher, Dalia Evenor, Eduard Belausov, Hanita Zemach, Adi Faigenboim, Samuel Bocobza, Michael P. Timko, Moshe Reuveni

**Affiliations:** 1Institute of Plant Sciences, ARO Volcani Center, P.O. Box 15159, Rishon LeZion 7528809, Israel; spatharaj@gmail.com (P.S.); yaarit@volcani.agri.gov.il (Y.K.); vhevenor@volcani.agri.gov.il (D.E.); eddy@volcani.agri.gov.il (E.B.); hanita@volcani.agri.gov.il (H.Z.); adif@volcani.agri.gov.il (A.F.); bocobza@volcani.agri.gov.il (S.B.); 2Department of Biology, University of Virginia, Charlottesville, VA 22904, USA; mpt9g@virginia.edu

**Keywords:** shoot regeneration, tobacco, regeneration induction

## Abstract

Shoot regeneration is a key tool of modern plant biotechnology. While many researchers use this process empirically, very little is known about the early molecular genetic factors and signaling events that lead to shoot regeneration. Using tobacco as a model system, we found that the inductive events required for shoot regeneration occur in the first 4–5 days following incubation on regeneration medium. Leaf segments placed on regeneration medium did not produce shoots if removed from the medium before four days indicating this time frame is crucial for the induction of shoot regeneration. Leaf segments placed on regeneration medium for longer than five days maintain the capacity to produce shoots when removed from the regeneration medium. Analysis of gene expression during the early days of incubation on regeneration medium revealed many changes occurring with no single expression pattern evident among major gene families previously implicated in developmental processes. For example, expression of Knotted gene family members increased during the induction period, whereas transcription factors from the Wuschel gene family were unaltered during shoot induction. Expression levels of genes involved in cell cycle regulation increased steadily on regeneration medium while expression of NAC genes varied. No obvious possible candidate genes or developmental processes could be identified as a target for the early events (first few days) in the induction of shoot regeneration. On the other hand, observations during the early stages of regeneration pointed out that regeneration does not occur from a single cell but a group of cells. We observed that while cell division starts just as leaf segments are placed on regeneration medium, only a group of cells could become shoot primordia. Still, these primordia are not identifiable during the first days.

## 1. Introduction

In 2005, the question “How does a single somatic cell become a whole plant?” was among the 25 most important questions in the next quarter-century of biology [1]. Somatic regeneration or embryogenesis is a synthetic process where a plant shoot or embryo (embryogenesis) or a root (rhizogenesis) is formed from a single somatic cell. What remains to be determined is whether the reprogramming of somatic cells into a new shoot or root is truly embryogenesis since only a partial plant is formed. Furthermore, whether this is a direct embryogenesis from a single cell or indirect process from a multicellular origin.

Plants have a remarkable developmental capacity to replicate via somatic cells without fertilization [2,3], leading to clonal proliferation. Steward et al. [4] showed that segments of mature carrot tissue regenerated whole plants [4], demonstrating the existence of totipotent plant somatic cells. In various plant tissues, regeneration occurs in response to applying exogenously applied phytohormones and plant growth regulators (PGRs), with auxins and cytokinins being the most significant. Somatic shoots can arise from several differentiated tissues in response to exogenous or endogenous stimuli [5,6]. Thus, somatic embryos and shoots are a powerful biotechnological tool for plant propagation and genetic improvement.

The ratio of externally applied phytohormones or PGRs determines the types of cells developed from the somatic tissue. Usually, a high cytokinin to auxin ratio results in shoots, while a low cytokinin to auxin ratio results in roots development. Intermediate cytokinin to auxin ratios form a callus. The regeneration of shoots and roots is divided into three stages; competence, induction, and development [7]. Competence is the acquisition of the ability to regenerate (i.e., the tissues acquire the ability to respond to growth- regulators) and, thus, shoots or roots can be induced. The induction stage is the initiation of regeneration where the developmental fate of competent cells is determined (i.e., shoot, roots, or callus is formed from the induced tissue evoking the cells totipotent nature). The third stage is developing the induced organs determined in the second stage (i.e., shoots, roots, callus, or embryos under the regular developmental program of meristems). Ectopic expression of meristematic and embryonic genes (e.g., Shoot Meristemless (STM), Baby Boom (BBM), Enhancer of Shoot Regeneration (ESR1 and ESR2), Leafy Cotyledon (LEC), etc.) bypasses the early stages and forces somatic cells to produce shoots [8].

The molecular events that lead somatic cells from leaf tissue to form a new shoot in tissue culture are not very well understood. Reprogramming competent somatic cells into totipotent cells is the initial step in somatic shoot formation and embryogenesis [9].

Genetic analysis of the “shoot regeneration” trait in which the F_1_ and F_2_ progeny are analyzed for the “shoot regeneration” phenotype is based on the assumption that the Quantitiative Trait Loci (QTL) for the trait does not affect whole plant development. The fact that these QTL do not affect plant development is not a trivial assumption because it implies that the disturbance in the process of shoot regeneration in culture has no direct connection to normal plant development and embryogenesis, rather the disruption of genes that are thought to be involved in somatic embryogenesis [10]. In other words, when plants are recalcitrant to regeneration, genes that are triggered during shoot regeneration or somatic embryogenesis, or at least during the initial stages of shoot regeneration before normal development occurs, are not part of normal plant development. It is possible that the genes that affect shoot regeneration could be redundant members of gene families that are activated by growth regulators, such as in the case of the cytokinin response gene family [11]. These genes could be redundant genes from gene families involved in plant development, such as the family of KNOX genes [12], WOX genes [13], and NAC genes. Therefore, the spotlight should be on gene families and processes that may not be involved in normal development.

This supposition is also supported by empirical, experimental data showing that regeneration requires both auxin and cytokinin [14]. The crosstalk between auxin and cytokinin and the genes affected by this crosstalk are good candidates for involvement in the induction of shoot regeneration. Plant cells are somehow able to measure both the concentration of auxin and cytokinin and the ratio between the two [15,16]. Differential circulation of auxin within plant tissues or organs functions as the main signal for auxin-dependent plant developmental processes and, thus, subject to tight regulation. Polar auxin transport is a fundamentally important regulatory process of auxin signaling [17]. All of these information streams are then somehow integrated leading to the induction of shoot regeneration in the competent tissue.

Nitric oxide (NO) regulates growth processes such as vegetative and generative development, seed germination, root growth, gravitropism, flowering, and fruit ripening [18]. The growth regulating effect of NO is caused by auxin–NO interplay and cytokinins interaction, regulating cell division [19] and shoot regeneration [20]. Additionally, NO participates in the abiotic stress responses of plants [18].

Here we describe a model system that is simple, reversible, and has a high-frequency shoot regeneration that allowed us to study the early events of shoot embryogenesis and shoot regeneration in cultured tobacco leaf segments at the cellular and transcriptomic level. These analyses show that no single factor determines the regeneration commitment of competent tobacco leaf segments.

## 2. Materials and Methods

### 2.1. Seed Sterilization and Plant Growth Conditions

Seeds of tobacco (*Nicotiana tabacum* L. cv. SR1) were placed in a 1.7 mL micro-tube (Eppendorf) filled with sodium hypochlorite (0.5% active material) and incubated for 5 min. The tube was shaken during the sterilization. After incubation, the seeds were rinsed three times with sterile water and spread on Petri dishes contained ½-strength MS medium (Duchefa Co., Haarlem, The Netherlands, Product number M0221.0050). After about two weeks, seedlings were planted in polypropylene Vitro Vent containers (Duchefa, NL; 9.6 cm × 9.6 cm and 9 cm in height) containing the same media to obtain disinfected plants. Plants were grown in sterile boxes in a growth room with 16 h of light and 8 h of darkness at 26 °C for several weeks until leaves were ready to be harvested.

### 2.2. Leaves Preparation and Regeneration

Leaves were detached from clean plants, and the midrib was removed. The leaf blade was cut into about 25 mm^2^ (5 mm × 5 mm) segments and placed on regeneration (Reg) medium containing standard MS salts according to Evenor et al. [15,16], supplemented with 30 g l^−1^ sucrose and 8 g L^−1^ agar, and the following growth regulators: 4.57 μΜ IAA; 9.29 μΜ Kinetin, and 4.56 μM Zeatin (all from Duchefa Co.). The control medium was the same but lacking growth regulators. The medium was adjusted to pH 5.6. At least 20 leaf segments were placed on each Petri dish with at least four plates per treatment in all experiments. When GUS (ß-glucuronidase) activity was tested, leaf segments were incubated in GUS activity buffer described before by Evenor et al. [16].

The effect of pre-incubation on Reg medium was tested by placing leaf segments on Reg medium for 1 to 8 days, with samples being move to MS medium after each indicated time period. Regeneration of leaf segments was scored 30–32 days after placing them on the MS medium.

Analyses of variance (ANOVA) was performed with the SAS/JMP software (SAS Institute Inc., Cary, NC, USA). Differences among means were calculated based on the Tukey–Kramer honestly significant difference (HSD) test for three or more treatments and T-test for two treatments [21,22].

### 2.3. Histological Studies of Leaf Segments

Leaf segments were removed from the medium and fixed in FAA mixture (5: 5: 90) of glacial acetic acid: formalin (40% *v*/*v*): ethanol (70% *v*/*v*). After fixation, the segments were dehydrated in stepwise ethanol series (30%, 50%, 70%, 90% (*v*/*v*) each for 24 h, followed by two 24 h incubations in fresh 100% ethanol) and embedded in paraffin. Paraffin sections (12 µm thick) were cut and then stained with Safranin and fast green FCF [23,24]. The thin sections were examined with a light microscope (Olympus BX50, 20 to 40 magnification).

### 2.4. RNA Preparation and Transcript Detection

RNA was isolated from leaf segment using PureLink midi kit (Invitrogen, Waltham, MA, USA) according to the manufacturer’s instructions and then treated with DNAse (Turbo DNA-free™, Ambion, Waltham, MA, USA). Single strand cDNA was prepared from total RNA using RevertAid^TM^ first-strand cDNA synthesis kit (Fermentas, Tel Aviv, Israel). PCR was conducted on cDNA samples isolated from each day using primers shown below [21]. Microarray analysis using the total RNA preparations described above was carried out by a commercial vendor (Rouch-Nimblegen, Madison, WI, USA) using a proprietary microarray chip based on the TOBFAC tobacco gene database (http://compsysbio.achs.virginia.edu/tobfac/). The RNA was hybridized to the custom microarray to identify genes whose expression changes during the seven days of the shoot induction period. RNA was isolated from day 0, day one, day four, and day seven after leaf segments were placed on Reg medium. The following comparisons were performed.

Treatment 1. 1 day in induction compared to 0 days in induction medium.

Treatment 2. 4 days in induction compared to 0 days in induction medium.

Treatment 3. 7 days in induction compared to 0 days in induction medium.

Each of the above hybridizations was conducted in three biological replications. cDNA was prepared from total RNA isolated and qPCR analysis was performed as described [22] using the following primers:

NtWusforward-GTACGAGGTGGACACCCACAAC; NtWusreverse-GCAGCAGCAGCAATAAGCCTC; NtKnotted1forward–GGACAACAACAACAATAATCCAC; NtKnotted1reverse-CTTCCTCTTCTTCATGAACTCC; NtKnotted2forward-GTGAAGGCGTCGGATCGTCCGAAG; NtKnotted2reverse-CCCACCAACTGAGCAGCTTCTGGCGT; Nt18Sfor-GCGACGCATCATTCAAATTTC; Nt18Srev-TCCGGAATCGAACCCTAATTC; NtPDLP2for-ATTATCCTAATGGTGTGCCCG; NtPDLP2rev-AGCAACTCCTAAACCCACAC; NtPDLP3for-TCAGCACCAGATTACACTAAGTTAG; NtPDLP3rev-ACTTAGATTTTGAGGATTGTGCAAC; NtPDLP6for-GTTTATGACCAAATGCTACGCG; NtPDLP6rev-CTTTTCAACATCTTCATCTCCGC; SlANT1for-AGTTGTAGATTGAGGTGGCTGA; SlANT1rev-CCGGGAAGTCTACCAGCAAT.

All PCR primers used throughout this study were designed based on the tobacco genome in Sol Genomics Network (Sol Genomics Network) and purchased from Hy Labs Ltd. (Rehovot, Israel). Real-Time PCR (qRT-PCR) analyses were done as described by Schreiber et al. [22]. The PCR reactions were performed in a T-GRADIENT thermal cycler (Biometra, Analytik Jena, and Gottingen, Germany).

Total RNA was extracted from sepals and flowers of tobacco using the TRIzol reagent system (Invitrogen Corp, Carlsbad, CA, USA). Genomic DNA contaminants were digested with TURBO DNA-free DNAase (Ambion Inc., Austin, TX, USA). The remaining RNA was then used as the template for cDNA synthesis using the Masterscript cDNA synthesis kit with random hexamer primers (KAPA Biosystems, Woburn, MA, USA).

The qRT-PCR analysis was performed using the KAPA SYBER FAST Master Mix (KAPA Biosystems, Woburn, MA, USA). DNA sequences complementary to tobacco *18S RIBOSOMAL RNA* were used as control. Three technical replicates were performed for every biological repeat, three biological repeats for each condition. The qRT-PCR analyses were done using the Rotor-Gene Q detection system analyzed with the Rotor-Gene 6000 software (Qiagen Corbett Life Science, Dusseldorf, Germany). The relative abundance of the examined gene transcripts was calculated by the formula: 2^(CT_examined gene-CT_reference gene)^, where CT represents the fractional cycle number at which the fluorescence crosses a fixed threshold.

### 2.5. Data Processing and Digital Tag Profiling

By removing 3′ adaptor fragments and several types of impurities from the raw reads, we obtained clean sequences. Then, sequences were mapped to the tobacco genome sequences in the Sol genomics database. No more than two mismatches were allowed in the alignment [25]. A rigorous algorithm described previously [26] was used for statistical analysis to identify differentially expressed genes (DEGs). We determined each gene expression level by the reads number uniquely mapped to the specific gene and the total number of uniquely mapped reads in the library. The threshold P-values were adjusted by the multiple testing procedures described by Benjamini and Yekutieli [27] by controlling false discovery rate (FDR). In this study, FDR ≤ 0.01 and the absolute value of |log2Ratio|≥ 2 were used as the threshold for judging the gene expression significance. The DEGs were subjected to Gene Ontology (GO) database (http://www.geneontology.org/) and mapped to the reference canonical pathways in the Kyoto Encyclopedia of Genes and Genomes (KEGG). The target sequences were allocated to the corresponding functional categories based on the BLAST searches by GO annotation using default parameters. The gene expression patterns of each pairwise comparison (0_DAP-vs-1_DAP, 0_DAP-vs-4_DAP, 0_DAP-vs-7_DAP, 4_DAP-vs-7_DAP, 0_DAP-vs-7_DAP) was analyzed, and genes were clustered according to their expression level using a self-organizing map using Cluster 3.0 with all the default parameters except the Euclidean distance of similarity metric. Additionally, the expression values were log2-transformed. Heat-maps with cluster data were then constructed using Java Tree View (http://jtreeview.sourceforge.net/) for visualization of the hierarchical clustering results [28].

### 2.6. Chlorophyll and Segment Size and NO Measurements

Chlorophyll content and chloroplast numbers were measured according to Kolotilin et al. [29]. Each time point was the average of five replicas. The area of leaf segments on various media was measured using ImageJ free software [30]. Each time point is the average of 10 measurements ± SE.

NO staining at the border of leaf segments placed on regeneration medium was done by incubating the leaf segments in 50 mM MES buffer, pH 5.6, and 1 mM DAF-2DA (4,5-diaminofluorescein-2 diacetate) for 30 min prior to visualization. Stain intensity per cell was analyzed by using ImageJ free software [26]. The effect of NO on regeneration was done by placing leaf segments on Reg medium containing NO donors (20 μM, S-Nitroso-N-acetyl-DL-penicillamine; 5 μM, Molsidomine), NO scavenger (100 μM, PTIO), and NO synthesis inhibitor (5 μM, Diphenyleneiodonium)**.**

### 2.7. Tobacco Plant Transformation and Measurements of GUS Activity

Tobacco (*Nicotiana tabacum* cv. SR1) plants were grown in sterile vessels and leaf segments excised as needed for transformation. Transgenic tobacco plants expressing a reporter gene for cytokinin (AtARR5::GUS or Aux/IAA3::GUS) that was a gift from Prof. Joe Kieber (The University of North Carolina at Chapel Hill) and 35S::PDLP5-GFP (a gift from Dr Jung-Youn Lee, University of Delaware) [31] were generated via Agrobacterium-mediated transformation using standard cocultivation methods. Transgenic seedlings were grown to flowering, self-fertilized, and homozygous T2 plants expressing Aux/IAA::GUS were used in auxin responsiveness assay by observing GUS activity in response to plant growth regulators. For chimera analysis, transgenic tissues were generated using a tomato *ScANT1* anthocyanin-inducing plasmid (a gift from Dr. I. Levin Volcani Center). A total of 84 (5 mm^2^) leaf segments were transformed and each analyzed for the presence of the colored cells.

## 3. Results

### 3.1. Induction of Shoot Regeneration Occurs after a Few Days on Regeneration Medium

The process of shoot induction in tobacco tissue culture involves specific changes in the tissue’s responsiveness to auxin and cytokinin. When incubated on medium containing high cytokinin to auxin ratios (shoot regeneration), shoots are formed, and when plated on medium with low cytokinin to auxin ratio (root regeneration), roots are formed (Figure 1A). When leaf segments are placed on regeneration (Reg) medium for one to eight days and then transferred to phytohormone-free medium, shoots are formed after 4 to 5 days (Figure 1B). The induction period for shoot regeneration from leaf segments exposed to high cytokinin to auxin ratio was observed to be up to 4 days. Exposing the leaf segments to the Reg medium for more than 4–5 days was sufficient to induce high shoot regeneration levels (Figure 1B). After 8 days on regeneration medium, all segments regenerated shoots even though they were no longer exposed to growth regulators.

We also observed that during the first week on either medium, leaf-segments lost their chlorophyll very quickly (Figure 1C). There was no significant difference in chlorophyll degradation between leaf segments on MS or Reg medium during the first week (Figure 1C). However, 10 days after the segments were placed on Reg medium chlorophyll content started to increase. In contrast, the chlorophyll content of segments placed on MS medium did not increase. The area of the leaf segments on either medium showed similar increases in size during the first seven days (Figure 1D). However, after day seven on Reg medium, segments continued to gain size while segments on MS medium did not (Figure 1D). This observation may indicate that during the first seven days of incubation on Reg medium, both size gain and chlorophyll loss are not related to the medium’s growth regulators’ content. However, as shoot regenerations occur only in the Reg medium, shoot regeneration induction is linked to growth regulators’ presence.

### 3.2. No Multicellular Structures Are Observed during the Shoot Induction Period

Thin sections of leaf segments incubated from 1 to 8 days on Reg medium were examined over the course of the induction period. During the first days after placement on Reg medium (i.e., days 0 and 1), no change was observed in leaf cells below the cut area. The cytoplasm in the cells is very thin, there is no decrease in chloroplast numbers, and most of the cellular volume is occupied by a large vacuole. As the segment stays on the Reg medium, we observed the cytoplasm thickening (Figure 2). The numbers of chloroplasts showed a decline in parallel with the observed decrease in chlorophyll content (Figure 1C). While at day 0 and day 1, the cytoplasm is very thin, thickening of the cytoplasm starts to be visible at days 2 and 3, indicating that some cells are entering into mitosis [32]. Cell divisions are more visible at days 4 and 5 and can be observed more readily as small cellular clusters (Figure 2, short arrows). On days 6 and 7, bigger clumps of cells are observed, and on day 8, pre-shoot structures can be observed (Figure 2, long arrows). The cell clumps appear more organized on day 8 and pre-shoots structures can be distinguished. During the shoot induction period (days 1 to 5), no structures are observed except for cell divisions. The above observations indicated that during the induction period of about 7 days on Reg medium, cell division occurs very rapidly. Still, no meristematic of embryonic structures are visible, although there is a commitment to producing shoots that will be seen sometime in the future.

### 3.3. Shoot Regeneration Is a Multicellular Event

Transgenic plants occasionally show chimeric phenotypes resulting from only some of the cells in the tissue having acquired the transgene or a localized gene silencing event (Figure 3A–D). Careful examination of the regeneration pattern in fully-grown and stably-transformed T_0_ tobacco plantlets and plants (Figure 3) suggested to us that the chimeras are the result of the way shoots are formed during regeneration. PCR analysis for the presence of the *ScANT1* transgene in DNA isolated from red and green sections of the leaves (see Figure 3) indicated that the red parts contained an amplicon consistent with the presence of the transgene whereas the green parts of the leaves (panels E and G) show a band while those depicted in panels F, H, I do not have a band (Figure 3J). This observation suggests that the green areas are the result of transgene silencing in some cases and lack of transgene (i.e., true chimeras) in others (Figure 3J).

Leaf segments from transgenic plants expressing the 35S::ScANT1 gene construct were used to follow the fate of transformed (colored) cells during 45 days on a regeneration medium without with and without kanamycin selection to see if shoots are formed from a single cell (green or pigmented) in the absence of selection pressure to favor transformed cells (Figure 4). We transformed 84 segments and analyzed 230 transformation events (2.7 ± 0.3 pigmented cells per segment). All of the segments regenerated green shoots totaling 568 shoots (6.8 ± 0.4 shoots per segment). Eight shoots had anthocyanin containing parts (1.4% chimeras) and no red plants were observed. Leaf segments showing pigmented cells were observed 6 days after transformation (Figure 4A). After 9 days, cells proliferated, and pigmented clusters appeared (Figure 4B). Twenty-five days after transformation, pigmented sections were visible in the plantlets formed, and 45 days post-transformation, plants showed pigmented leaves as part of a green plant (Figure 4C,D).

### 3.4. Role of Cellular Signaling during Induction of Shoot Regeneration

We tested the effects of Nitric Oxide (NO) on shoot regeneration in tobacco leaf segments using different chemical treatments. A decrease in NO intensity was visible upon treatment with Diphenyleneiodonium (a known NO synthase inhibitor). In contrast, an increase in NO intensity was observed upon treatment with Molsidomine, an NO donor (Figure 5A,B). NO donors and inhibitors applied to leaf segments affected shoot regeneration. No effect on shoot regeneration was found following treatment of leaf segments with 2-phenyl-4, 4, 5, 5,-tetramethylimidazoline-1-oxyl 3-oxide (PTIO), a stable radical scavenger for nitric oxide that does not affect NO synthase. In contrast, Diphenyleneiodonium (Diphenyl), a potent and reversible inhibitor of NO synthase, has a significant inhibitory effect on shoot regeneration (Figure 5C). The NO donor S-Nitroso-N-acetyl-DL-penicillamine (SNAP) did not affect the number of shoots regenerated per leaf segments. In contrast, treatment with the NO donor Molsido (Molsidomine) increased the number of shoots regenerated per leaf segments (Figure 5C). Furthermore, the increase in NO levels during shoot induction was correlated with an increase in *NITRIC OXIDE SYNTHASE 1* transcript levels. In contrast, the level of *NITRIC OXIDE SYNTHASE 2* transcripts remained unaltered (Figure 5D) while the transcript levels of *NITRATE REDUCTASE* 1, 2, and 3 increased and then decreased in the induction period without any difference between the various alleles (Figure 5E).

Plasmodesmata (PD) are ubiquitous in plants and used as a means of cell-to-cell communications. We studied the development of plasmodesmatal connections in new cells that grow during the regeneration induction period (0 to 7 days on Reg medium). Leaf segments from transgenic tobacco plants expressing the 35S::AtPDLP5-GFP transgene (a protein that is part of the plasmodesmata structure, [31]) were placed on Reg medium, and PD in the new cells were visualized and analyzed (Figure 6A). The number of PD connections per cell increases from day 4 onwards (Figure 6B), while the fraction of cells with connections remain constant after day 6 (Figure 6B). Analysis of tobacco PDLP genes (NtPDLP2, NtPDLP3, and NtPDLP6) show that the tobacco PD genes start to increase from days four on Reg medium (Figure 6C).

### 3.5. Global Transcript Levels of Genes during the First Seven Days on Regeneration Medium

Transcriptional changes taking place during the shoot induction period in tobacco leaf segments incubated on Reg medium were analyzed using RNA samples collected at 0, 1, 4, and 7 days of incubation. A hybridization scheme was designed to identify genes whose expression changed specifically during the shoot induction period of 4 days and to eliminate genes constitutively expressed during the seven days of incubation on the regeneration medium. For this analysis we set a False Discovery Rate (FDR) cut-off value of 0.01 as the threshold for judging significantly different gene expression. As shown in Appendix A, hundreds to thousands of transcripts were differentially regulated during the shoot induction period (Appendix A). The multitude of transcripts can be divided into five subclusters according to their expression pattern (Appendix A). The transcripts that are up or down-regulated during the shoot induction period are not part of a particular biological process, but many biological processes seemed involved, at least at the RNA transcript level (Figure 7 and Appendix A).

Although it was not possible to define exactly which transcripts regulated shoot regeneration using a global analysis approach, it was possible to characterize the expression profile of specific meristematic genes. We identified 48 *Knotted*-type genes and examined their expression pattern during the shoot induction. As shown in Figure 8, members of the *Knotted* gene family showed variability in their pattern of expression during the shoot induction period. For example, *NtKnotted1* expression did not increase during the shoot induction period but only after day 10 when there is a visible meristem (Figure 8B). At the same time, NtKnotted2 expression increased on day 7 (Figure 8C), but was expressed overall at a lower level all times. Another large gene family in tobacco is the histone-related gene family with 64 members. Among the 64 histone-related gene family members expressed during the induction period, 33% did not change, 40% were upregulated at day 4 and remain high at day 7, and 27% peak at day 4 of the induction period and then decrease considerably at day 7 (Appendix A). The histone-related gene family pattern is indicative of active DNA metabolism and rearrangement (Appendix A).

One hundred and two genes were identified in the NAC gene family in tobacco [33,34], but only eight transcripts were identified with varied expression during the shoot induction period (Appendix A). Genes ID 1203313, EH666187, TC18954, and TC23573 increased 30-, 20-, 16-, and 8-fold on day 4 and then dropped on day 7 while genes ID 5124 and 1129698 increased 22-fold and 14-fold, respectively, on day 1 and then fell to 2-fold on day 4 and day 7. Gene ID BP527214 increased 6-5 fold on all days tested compared to day 0, while gene ID 5153 decreased 2.5 fold on day 7 (Appendix A).

Interestingly all members of the MADS-box gene family that changed significantly after day 0 did not show an expression pattern alteration on the regeneration medium during the timeframe examined (Appendix A). Wuschel genes (WUS or WOX genes) are well-known meristematic maintenance genes thought to be expressed only in the meristem [35,36]. Wuschel genes are linked to shoot regeneration and embryogenesis [35]. No obvious pattern was found in common among the members of this gene family, with some Wuschel gene family members increasing during the shoot induction period (Figure 9A,B), whereas others decreased (Figure 9A).

### 3.6. The Positional Effect of Leaf Segments Affects Shoot Regeneration

When tobacco leaf segments are placed on Reg medium and the agar plates are inverted such that the leaf segment is beneath the agar (i.e., the segments were placed on the agar, and the plates were turned upside down), there is a decrease in both percent regeneration and the number of regenerated shoots (Figure 10A,B). Rotating the plates show similar regeneration as non-turned plates (Figure 10A,B), implying that gravity changes do not affect regeneration. We next tested leaf segments from tobacco plants that were transformed with reporter genes for auxin (IAA/AUX3::GUS) and cytokinin (ARR5::GUS). There was no difference in cytokinin responsiveness in the leaf segment at early stages when placed on media, but 30 days later, all segments showed GUS activity regardless of treatment (Figure 1A). Leaf segments containing the auxin reporter transgene were places on Reg medium and showed a significant increase at days 2 to 3 (Figure 10C). However, the responsiveness of the segments that were placed beneath the agar (down position) was reduced at days 2 to 3 compared to when the segments were placed on top of the agar (up position) (Figure 10C).

## 4. Discussion

The above findings indicate the interplay among plant growth regulators and the complexity of the interactions involved in shoot regeneration. Placing tobacco leaf segments on a regeneration medium for up to 4 days and removing them to a medium without growth regulators does not lead to *de novo* shoot regeneration, a longer period on regeneration medium induced shoot formation. During the first week on regeneration medium, tobacco leaf segments lost chlorophyll and increased in segment area similarly to leaf segments on medium without growth regulators. Placing the tobacco leaf segment for a longer period leads to de novo shoot regeneration on medium without growth regulators, to a further rise in segment area and increased chlorophyll. After seven to 8 days on regeneration medium, all tobacco leaf segments will regenerate shoots. There are minor changes in segment morphology, mostly expansion and thickening. During those seven days, cross-sections of the leaf segments show cellular multiplications and cytosolic thickening, indicating increased mitosis [32]. However, after more than five days on Reg medium, leaf segments will still develop shoots when removed from the Reg medium and placed on medium without plant growth regulators. This observation indicated that exposure to growth regulators in the Reg medium for 5–7 days initiates a nonreversible chain of events that will result in the formation of a shoot some 20–30 days later in the absence of plant growth regulators. Exposure to growth regulators up to 4 days is reversible, indicating that although leaf cells are competent to produce de novo shoots, there is an induction period in which leaf cells proliferate but are not committed to regeneration. This induction period indicated that some other factors are needed to induce shoot regeneration and that these factors are activated after four days in the presence of plant growth regulators.

Regeneration of de novo shoots requires both auxin and cytokinin [14], and the crosstalk between them is considered essential for the induction of shoot regeneration [15,16]. As we show above, plant cells must measure both the concentration of auxin and cytokinin and the ratio between them to decide whether to produce shoots roots or callus [15,16]. Polar auxin transport is a fundamental important regulatory process of auxin signaling [17]. Thus, the differential gradient of auxin within the plant tissues or organs can function as the main signal for auxin-dependent plant developmental processes and, therefore, subject to tight regulation. For appropriate tissue formation during development, cells must know their position relative to other cells. In animals, it is well documented that gradients of signaling molecules (morphogens) that convey positional information determine cell fate [37]. Positional information is remarkably missing from plant tissue culture studies as explants are always placed on agar plates or liquid floats or in liquid media. When we tested for positional information of the leaf segment, we found that there is a peak in auxin responsiveness before commitment to regeneration occurs. This peak in auxin responsiveness coincides with the start of plasmodesmata production and deposition in the cell wall.

Another cellular messenger molecule is NO that regulates growth processes such as vegetative and generative development, seed germination, root growth, gravitropism, flowering, and fruit ripening [18]. NO also participates in the abiotic stress responses of plants [18]. The growth regulating effect of NO is caused by the auxin–NO interplay and cytokinin interaction, regulating cell division [19] and shoot regeneration [20]. Our data show that NO has an important role in shoot regeneration and that inhibiting NO production inhibits shoot regeneration. All the information that the leaf cells receive from starting to communicate with their neighbors is then somehow translated to induction of shoot regeneration in the competent tissue.

During the induction of shoot regeneration, cells in close proximity to each other divide and start exchanging information that triggers the differentiation and commitment into a meristem and eventually a shoot. Thus, we wanted to determine whether global changes in gene expression or changes in the expression of particular genes occur during this induction period without the complication of developmental genes. Meristematic genes (WUS or WOX and Knotted or KNOX gene families) are part of the maintenance genes linked to shoot regeneration and embryogenesis [36]. Some of these genes increase during the shoot induction period, and some decrease. The altered expression of meristematic genes starts at day one on the regeneration medium and continues through the induction period. *WUSCHEL* is linked to shoot regeneration [35,36] and our data show that increasing *WUSCHEL* expression in the first four days of the shoot induction period is not sufficient to induce shoot regeneration since removal from Reg media during those four days reverses the process. Ectopic expression of *STM* and *WUSCHEL* activated a subset of meristem functions, including cell division, and ultimately organogenesis [38]. This overexpression suggests that *WUSCHEL*, combined with STM, initiates certain leaf cells’ progression to organ initiation [38]. Global gene expression of particular genes of interest analyses could not pin down a single or specific process necessary to induce shoot regeneration. This observation that a multitude of genes change when global gene expression is studied during regeneration was shown in Arabidopsis as well [8]. All the gene families check in this study show a similar pattern. Several gene family members increase during the induction period and several decreases, and most do not change. It seems that shoot regeneration is a complex phenomenon that involves many processes and is not triggered by a single gene. There was no single process within the cells regenerating into a shoot at the induction period.

Transformation of tobacco with *SlANT1* (a tomato MYB transcription factor that induces anthocyanin synthesis in tomato and tobacco plants) [18] showed visual chimerism. Using this phenotype, we monitored the process of shoot regeneration. We observed that without selection pressure, tobacco shoots are formed from a cluster of cells that do not originate from a single cell, thus, leading to chimerism. Chimerism in transformed plants can explain variations in transgene expression of the regenerated plant as well as the non-mendelian distribution of transgenic progeny that occurs sometimes. It seems that the way to overcome the chimerism problem is using at least T1 seeds and in targets that do not produce seeds, be aware that chimerism can obscure the results.

The process of tissue regeneration is present in most multicellular organisms but is restricted to certain organs or tissues in different organisms [39]. In most cases, wounding is required to induce regeneration in all plants and animals. Compared to animals, plants retain a high degree of developmental plasticity and display various tissues or organ regeneration [40]. Regeneration can range from repairing a small wound or amputation to new organs or individual organisms and varies markedly between taxa. The ability of cells to determine their position in space and communicate with their neighbors is crucial to establishing proper patterning in a developing or regenerating organ. While no stem cell movement is present in plants, we show that auxin flow direction is a determinant in shoot regeneration. Thus, cells’ capability to establish their environment and position is essential to the regeneration process through taxa [37]. It was shown that with the positional change of the root cells’ fate, the chromatin state is remodeled [41]. Thus, positional information and phytochemical crosstalk is essential to plant cell regeneration, not because each cell needs to know its environment, but because plant cells regenerate as a group.

The ability to clonally propagate various plants, modify plants by introducing foreign genes (like Bt protein), or modify existing genes by genomic editing in crop plants, is an important component of modern agricultural biotechnology. The plasticity of plant cells and, thus, tissue regeneration, is paramount to all these current bio-agricultural industries. Understanding the regeneration process can enhance the ability to control various aspects of tissue regeneration, making the process more efficient and target oriented. Our findings indicate that leaf tissue regeneration requires positional information between the cells and their cellular neighbors. What is unknown is how they are connected, and the signals they exchange. Modifying or controlling these parameters can enhance or decrease tissue regeneration. For example, adding NO enhances shoot regeneration from leaf segment.

Our findings indicate to complexity and the interplay among plant growth regulators and the interactions involved in shoot regeneration. During the very early stages of regeneration cell divisions and cellular communication is essential for regeneration that thereafter occurs from not from a single cell but a group of cells. We observed cell divisions just as leaf segments are placed on regeneration medium, only groups of cells become shoot primordia but these are not discernable during the first days and early stages of regeneration.

## Figures and Tables

**Figure 1 plants-10-00058-f001:**
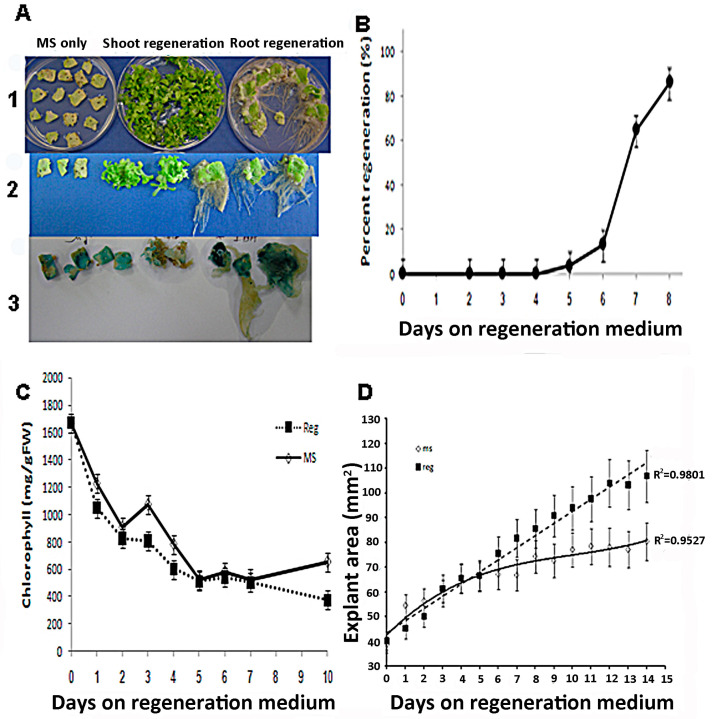
Tobacco leaf segments regeneration. (**A**1) Tobacco leaf segments were placed on MS nutrient medium or Regeneration medium, including auxin and cytokinin or MS medium (see Materials and Methods) containing auxin only (4.44 M IAA). Thirty days after placement on a specific medium, the segments were pictured. (**A**2) Close-up of leaf segments from the plates in (**A**1). (**A**3) Leaf segments from transgenic tobacco expressing a cytokinin reporter gene ARR5::GUS show responsiveness to cytokinin 30 days on the respective media described in (**A**1). (**B**) Time on regeneration medium that is required to induce shoots from competent tobacco leaf segments. Leaf segments were placed on regeneration medium for the indicated time and then transferred to MS medium for 30 days. Percent of shoots forming leaf segment was recorded after 30 days on MS medium + SE. Each point represents three plates with 20 segments. (**C**) Chlorophyll content of leaf segments on MS or regeneration medium. Leaf segments were placed on agar plates with MS only (MS) or with added growth regulators (REG), and chlorophyll was extracted and determined from each segment. Chlorophyll content per gram fresh weight from leaf segment was recorded on ten segments every day + SE. (**D**) Area of leaf segments explants during the first two weeks on MS or REG medium. Leaf segments were placed on agar plates with MS salts only (MS) or with added growth regulators (REG), and the area of segments was recorded daily to measure cell proliferation. Size of leaf segment was recorded 20 segments every day on the two media + SE.

**Figure 2 plants-10-00058-f002:**
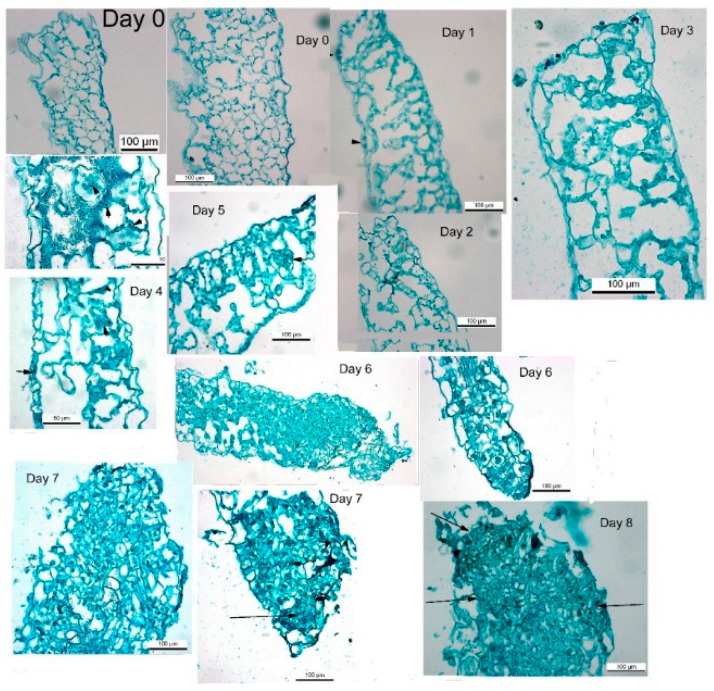
Cross sections of leaf segments taken during the first 8 days on REG medium. Arrowhead in Day 1 indicates stomata and arrows in Days 4, 5, 7, and 8 indicate cell clusters with cell divisions which will may form shoots.

**Figure 3 plants-10-00058-f003:**
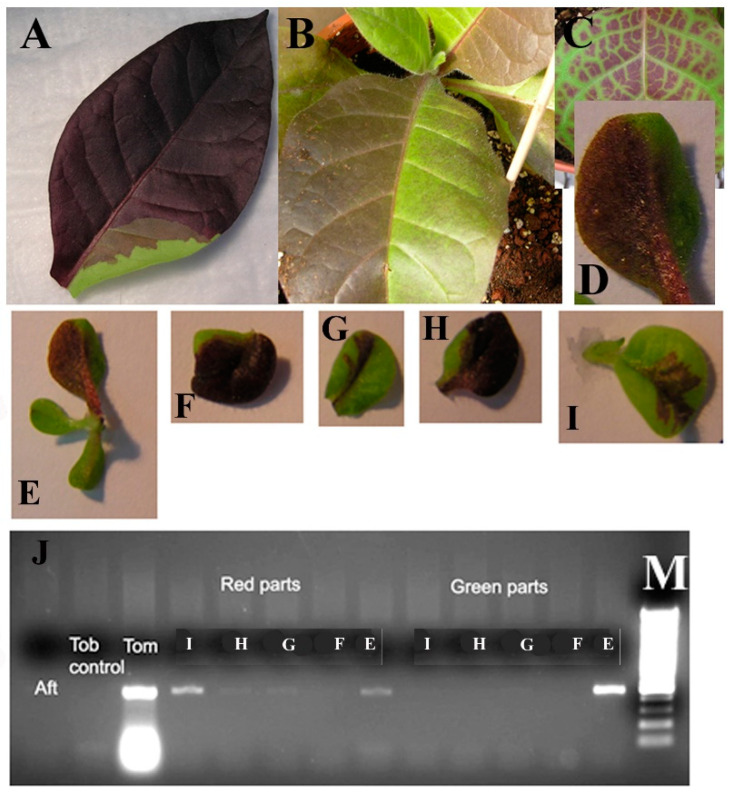
(**A**–**D**) Leaves of transformed tobacco show chimerism even on selection medium after transformation with tomato *ScANT1* gene construct and selection on Kanamycin. (**E**–**I**) Pictures of tobacco leaves transformed with the *ScANT1* construct. (**J**) PCR analysis of genomic DNA isolated from Red or Green sections of the leaves in (**E**–**I**).

**Figure 4 plants-10-00058-f004:**
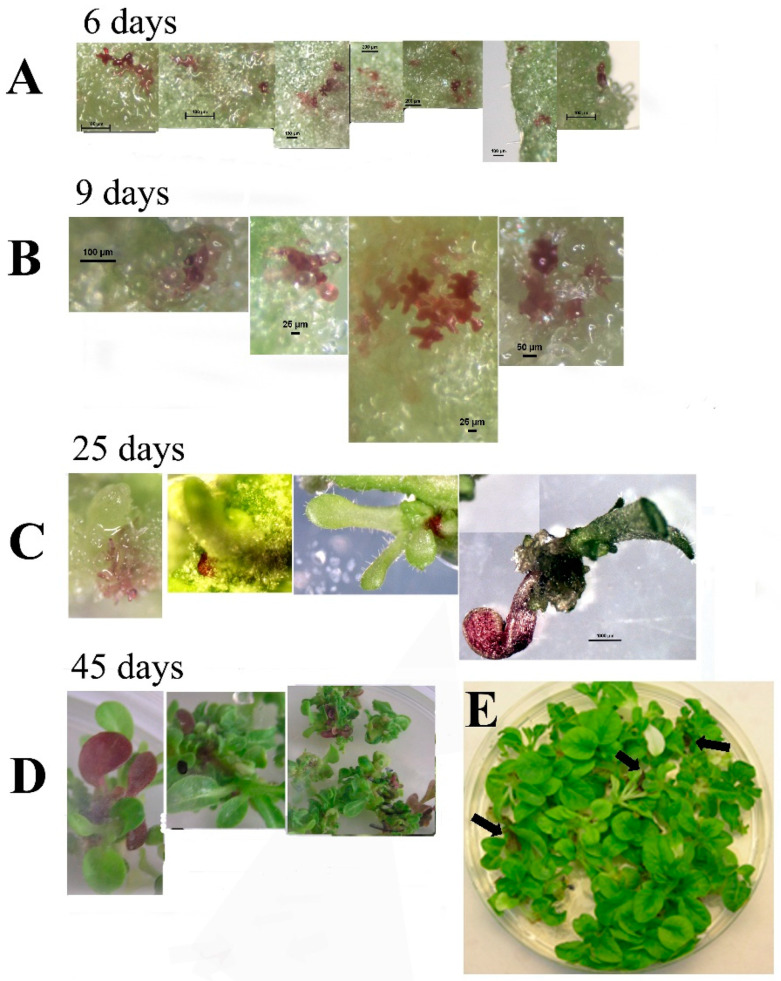
Bright light pictures of leaf segments transformed with the 35S::ScANT1 construct, taken at various time points following transformation and explant growth on regeneration medium without Kanamycin selection. (**A**) leaf segments showing pigmented cells 6 days after transformation. (**B**) leaf segments showing pigmented cells 9 days after transformation. (**C**) tissue showing pigmented section 25 days after transformation. (**D**,**E**) plants showing pigmented leave 45 days after transformation. Arrows point to the colored leaves in the plate.

**Figure 5 plants-10-00058-f005:**
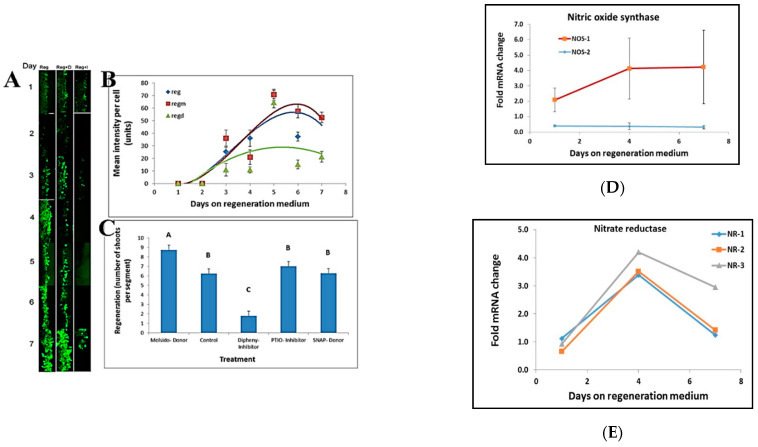
NO has a role in shoot regeneration. (**A**) NO staining at the border of leaf segments placed on regeneration medium in the presence of NO inhibitor (Reg+I), NO donor (Reg+D) and control (Reg). (**B**) Stain intensity per cell at the borders of leaf segments during the induction period of shoot induction. NO inhibitor (Regd), NO donor (Regm), and control (Reg). (**C**) Effect of NO donors (S-Nitroso-N-acetyl-DL-penicillamine; Molsidomine), NO scavenger (PTIO), and NO synthesis inhibitor (Diphenyleneiodonium) on shoot regeneration. Different letters (A, B, C) define statistically different results, p(f) was 7.65 × 10^−35^ using Tukey test. (**D**) Transcript level of Nitric Oxide Synthase1 increases during shoot induction while transcript level of Nitric Oxide Synthase2 is unchanged. The data presented are from the Differential Gene Expression experiment. (**E**) Transcript level of all Nitrate reductase genes increase during incubation on Reg medium. The data presented are from the Differential Gene Expression experiment. Reg = regeneration medium; regm = regeneration medium with added Molsidomine; regd = regeneration medium with added Diphenyleneiodonium. NR = nitrate reductase; NOS = nitric oxide synthase. Error bars indicated standard error of 50 cells in (**B**); 20 segment in 3 replicated in (**C**); and 3 biological replicates in (**D**,**E**).

**Figure 6 plants-10-00058-f006:**
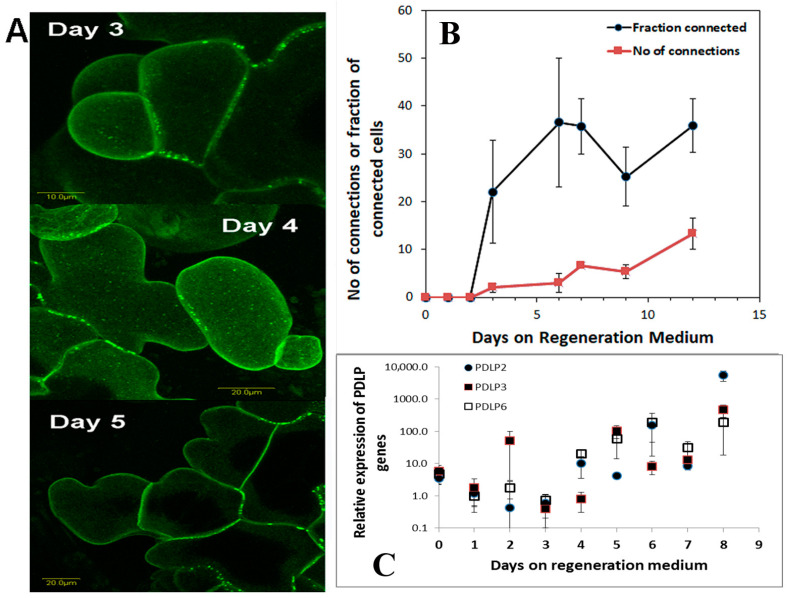
Plasmodesmata during shoot regeneration. (**A**) Leaf segment from tobacco plants that are homozygous to the transgene 35S::AtPDLP5-GFP were placed on Reg medium and samples were removed every day and the new cells were visualized and analyzed using confocal microscopy. (**B**) The number of PD connections per cell was counted and fraction of cells with PD from all new cells was counted. (**C**) Expression pattern of three tobacco PDLP genes during the shoot induction period is depicted. Error bars indicated standard error of 50 cells in B; and 3 biological replicates in (**C**).

**Figure 7 plants-10-00058-f007:**
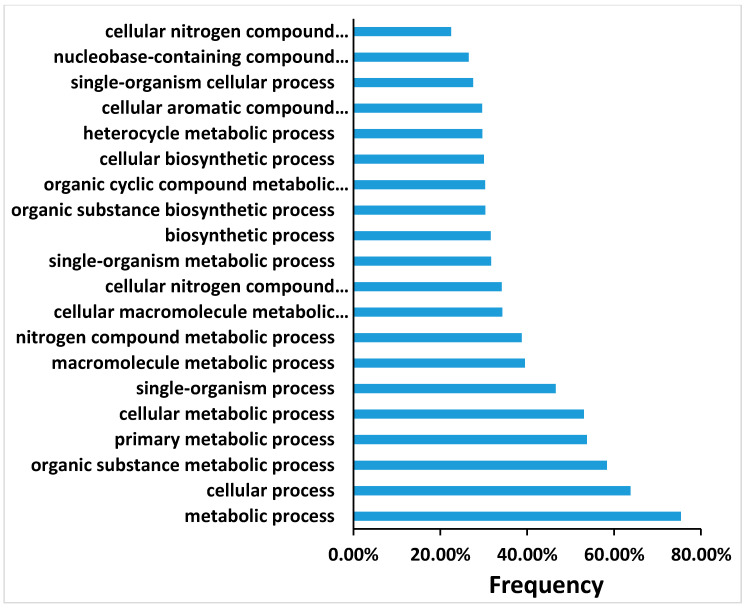
A sample of transcript analyses of top 20 biological processes that upregulate at day 4 of the induction of shoot regeneration. Only some of the processes were fitted on the Y-axis.

**Figure 8 plants-10-00058-f008:**
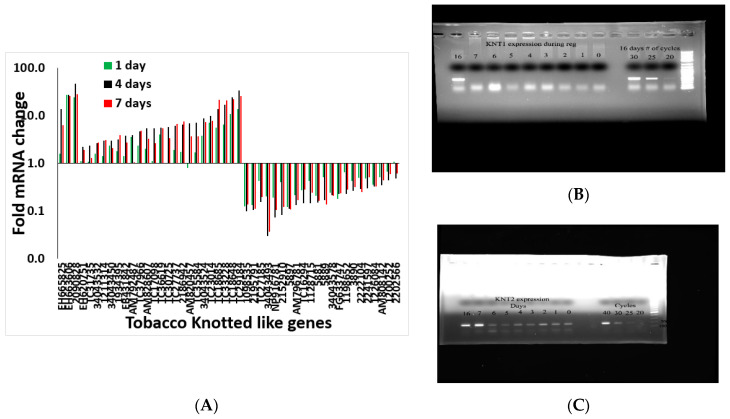
(**A**) Transcript pattern of Knotted gene family members during the induction period of shoot regeneration. The gene numbers are depicted on the x-axis. (**B**) expression of NtKnotted1 during the induction period. (**C**) expression of NtKnotted2 during the induction period.

**Figure 9 plants-10-00058-f009:**
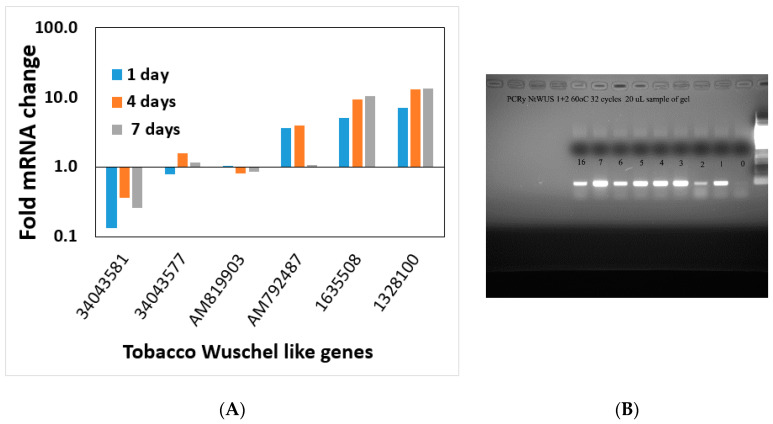
(**A**) Transcript pattern of Wuschel gene family (WOX) members during the induction period of shoot regeneration. The gene numbers are depicted on the x-axis. (**B**) Expression of NtWus during the shoot induction period on Reg medium.

**Figure 10 plants-10-00058-f010:**
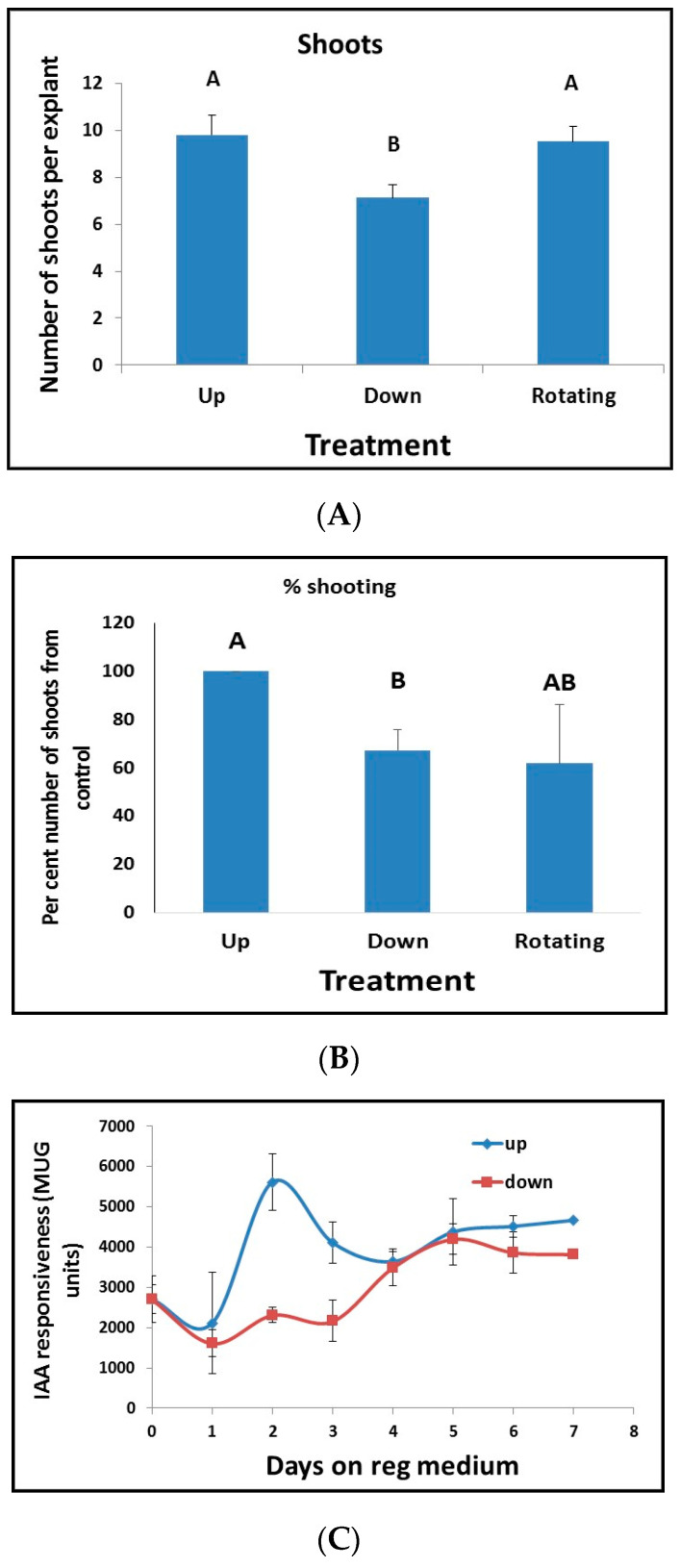
The orientation of leaf segment affects shoot regeneration and percent regeneration. (**A**) Number of shoots regenerated from leaf segments placed with the segment above the agar Reg medium (**Up**); leaf segments placed with the segment on the agar and the plates turned upside down, thus, the segment are beneath the agar Reg medium (**Down**) but still exposed to the air on one side or leaf segments placed with the segment above the agar Reg medium, and the plate was rotated at 30 rotations per min (**Rotating**). Results are the mean ± se of five repetitions representing 802 leaf segments. Different letters (A, B) define statistically different results (**B**) Percent regeneration from leaf segments placed as in (**A**), percent shooting was calculated as the number of segments that produced at least one shoot out of a total number of the segment in a plate. Results are the mean ± se of five repetitions for each treatment. Different letters (A, B) define statistically different results (**C**) Effect of positioning leaf segments containing the auxin reporter transgene IAA/AUX3::GUS above or beneath the agar medium on IAA responsiveness; up and down as in A. Each point is the mean ± se of five measurements of GUS activity done on ten individual leaf segments.

## Data Availability

All the data are in the manuscript.

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
