# Peer review of "Shoot Regeneration Is Not a Single Cell Event"

_plants, 2020, doi:10.3390/plants10010058_

Round 1

Reviewer 1 Report

1). Manuscript ID: Plants-1037967

2). Manuscript Title: Shoot regeneration is not a single cell event

3). General comments

--Extensive English changes required. Please edit the manuscript thoroughly before submission.

--Please follow the Journal format while revising the manuscript.

--Add scientific authority at the end of binomial names of all species when they are mentioned for the first time in the manuscript.

--Include full forms of all abbreviations/acronyms mentioned in the manuscript.

4). Specific comments

--Include high quality images for figure 1.

--Line 91: Provide culture tube dimensions in an appropriate format.

--Line 112: Dehydrated in stepwise ethanol? Please elaborate?

--Table 1: Mention the primer sequence direction.

--Line 161: ‘replicas’ or ‘replicates.

--Lines 386 to 388: Rewrite these lines.

Author Response

Reviewer 1

1). Manuscript ID: Plants-1037967

2). Manuscript Title: Shoot regeneration is not a single cell event

3). General comments

--Extensive English changes required. Please edit the manuscript thoroughly before submission.

The manuscript was revised by Prof. Mike Timko from University of Virginia.

--Please follow the Journal format while revising the manuscript.

We followed the Journal instructions

--Add scientific authority at the end of binomial names of all species when they are mentioned for the first time in the manuscript.

We added L to Nicotiana tabacum L.

--Include full forms of all abbreviations/acronyms mentioned in the manuscript.

We explained the abbreviations at first mention with the exception of gene names.

4). Specific comments

--Include high quality images for figure 1.

We modified Figure 1 according to both reveiwers sugenstions, increase the size of day0 pictures and the quality of the figure.

--Line 91: Provide culture tube dimensions in an appropriate format.

The text was amended to the right format.

--Line 112: Dehydrated in stepwise ethanol? Please elaborate?

Samples are placed in 30% ethanol for 24 h, then moves to 50% ethanol for 24 h, then moves to 70% ethanol for 24 h, then moves to 90% ethanol for 24 h, then moves to 100% ethanol for 24 h, then the 100% ethanol is replaced with a new batch of 100% ethanol or 24 h and ready for embedding.

--Table 1: Mention the primer sequence direction.

All the primers mentioned have an ending for (for forward) and rev (for reverse).

--Line 161: ‘replicas’ or ‘replicates.

We used in the manuscript replicates

--Lines 386 to 388: Rewrite these lines.

The lines were changed as follows: “From the 64 histone-related gene family transcripts that expressed during the induction period, 33% did not change, 40% upregulated at day four and remain high at day 7, and 27% decrease considerably at day 7 after peaking at day 4 of the induction period.  The histone-related gene family pattern is indicative of active DNA metabolism and rearrangement (Fig. S6).”

Reviewer 2 Report

Introduction:

Include information about all why you would look at Nitric Oxide and Positional effects.

Materials and Methods.

2.1 Provide more details of the MS media used, product number and company.

Check on abbreviation for hours and apply to whole of manuscript.

2.2

Line 103.  Detail the reference by name, don’t use the reference number.

2.4

Line 120-121 appear to say the same thing as Line 132.

Line 125.  Days should be day

This section is missing a lot of information.

  • PCR conditions
  • What is the house keeping gene and primers for the qPCR
  • pPCR details
  • Are primers designed to only amplify cDNA? i.e. do they cross introns.
  • What method was used for calculating the qPCR results, ΔΔCt?
  • How was the fold change calculated?
  • What are the reps, technical or biological? qPCR usually has 3 technical and you then have your biological reps.  Explain how you handled the results.
  • Primers for NO synthase genes and NO reductase genes are not detailed. Also explain how the primers discern between the different alleles.
  • Provide more information about where the sequence information came from for design of primers.
  • Were PCR and qPCR products sequenced to confirm correct gene amplified? Espcially for the allele expression results.

There are no materials and methods for the Nitric Oxide experiments or the Positional experiments or how choloroplast numbers were coutned.

Results

There are some references within the results.  Please confirm with the journal is this is acceptable.  The us of references shows points of discussion that may need to be moved to that section; unless is it confirming experimental protocol.

Line 208.  There is no data to say that chlorophyll content ‘continued to increase’.  Results are only shown for up to 10 days.

Line 218 – 222.  This appears to be a summary of the following paragraph, is it needed or should be moved to after the detailed paragraph?

Figure 2.  The images should be resized so that the scale bar is similar for all of them.  This will allow easier comparision between images.  i.e. cannot really compare day 0  to day 5 as the size of the day 0 image is very small.

Line 227.  How did you measure the chloropolast numbers?  (Details should be in Materials and Methods)

Line 242, Figure 3 should be moved to the end of the sentence.

Line 243.  What type of DNA?  gDNA would confirm the integration of the transformation event, cDNA would confirm transcription.

Figure 3.  The bands on the gell are very hard to see.  It appears that cDNA of the transgene was detected in the red leaf segements of plants I, H, G and E only, not F.  And in the green leaf segments for plants G and E only, not I, H and F.  Would it be possible to change the image to white and black instead of black and white, making the bands black.  Thank you for including the positive and negative controls.

Line 253.  Reword sentence to remove We.

Figure 4. There is no reference to 4E in the results, suggest removing it.  If incorporated the pigmented leaves may need to be highlighted with an arrow.

3.4

Line 279 and 282.  Figure 5C

Line 285.  The transcript levels increased and decreased, not just increased.

Figure 5.

Detail what the error bars are, standard error or standard deviation?  Include number of samples to calculate averages and error bars.

5B.  The mean intensity of what?  How is this measured?  What happened in Day 5?  The day 5 results significantly alter the inferred lines.  If day 5 was not included the inferred lines would look different. 

5C. What is the P level for the comparison?

Figure 6B.  Correct spelling of y-axis.

Figure 6. Detail what the error bars are, standard error or standard deviation?  Include number of samples to calculate averages and error bars.

Figure 7.  I appreciate that this is a sample of the transcript analysis, but the processess listed mean nothing as you don’t know what bar they represent.  Could you focus on the top 20 processes or the 10 highest/lowest processes and show their bars and include the full information in a supplementary figure/table.  This figure only shows that there was differences in expression profiles.

Line 338-342.  Data not shown for results presented.  Include in a supplementary table.

Line 343-348  Data not shown for results presented.  Include in a supplementary table.

Figure 8.  Fix figure, Day 4 and day 7 colours?  Where are the positive and negative controls for the PCR gel.

3.6

Need information about how plates were prepared.  Was the leaf segment placed on the agar and then turned upside down, or placed under the agar?  What was the lgiht conditions?  Could light be a factor, not gravity?

Figure 10.  Provide further detail as to how % shooting was calculated.  Maybe include values in a supplementary table.

Line 391. > 10 days.

Line 400-401.  Do you mean that after 4 days exposure the response is irreversible?

Lin 410. Different reference syle.

Discussion.

The discussion is more a summary of the results and does not show much comparison with published literature.  Published literature may be scarce. 

Why did you do the nitric oxide and positional experiments?  Was there other literature that suggested you should test these? 

How does the change in gene expression compare to other plant expression profiles? 

Any ideas how to ensure regenerated plants originate from single transformation events?

No discussion on the differences in expression of gene families.  A general statement with examples and correct references of how complex gene families are and how expression can be dependent on tissue/cell localisation, plant development stage, environmental conditions to name a few.

No discussion on the lack of specific biological processes being identified.  I would not expect a single process as within the clump of cells regenerating into a plantlet there are a multitude of cell types being formed. Unlike a matuer plant where leaf cells are clearly defined from stem cells.  There could be further discussion about this.

Comments for Editors (will not be shown to authors):

There has been an extensive set of experiments reported here to show that plantlet regeneration originates from a cluster of cells, both transformed and untransformed cells.  This is very important, especially when transgenic plants are being produced and results reported on those plants.  Variations in transgene expression may be due to the chimeric nature of the regenerated plant.

The authors have not fully discussed the results with respect to other published data and the discussion needs extensive work.

There are sections of the materials and methods missing that I have highlighted.  I have also suggested the use of more supplementary figures/tables so that readers can assess results themselves rather than having it reported to them.

Author Response

Manuscript ID: Plants-1037967

Reviewer 2

Comments and Suggestions for Authors

Introduction:

Include information about all why you would look at Nitric Oxide and Positional effects.

NO (Nitric oxide) regulates growth processes such as vegetative and generative development, seed germination, root growth, gravitropism, flowering, and fruit ripening (Kolbert 2019). Furthermore, NO also participates in the abiotic stress responses of plants (Kolbert 2019). The growth regulating effect of NO is caused by auxin-NO interplay and cytokinins interaction, regulating cell division (Shen et al. 2013) and shoot regeneration (Arun et al. 2017). 

Kolbert Z. Strigolactone-nitric oxide interplay in plants: The story has just begun. Physiologia Plantarum 165: 487–497. 2019

Shen Q, Wang YT, Tian H, Guo FQ (2013) Nitric oxide mediates cytokinin functions in cell proliferation and meristem maintenance in Arabidopsis. Mol Plant 6: 1214–1225.

Arun, M., Naing, A.H., Jeon, S.M., Ai, T.N., Aye, T., Kil Kim, C.K.  2017 Sodium Nitroprusside Stimulates Growth and Shoot Regeneration in Chrysanthemum. Hortic. Environ. Biotechnol. 58(1):78-84. 2017.

Differential circulation of auxin within plant tissues or organs functions as the main signal for auxin-dependent plant developmental processes and thus subject to tight regulation. Polar auxin transport is a fundamental important regulatory process of auxin signalling (Tanaka et al. 2006).

Tanaka, H.; Dhonukshe, P.; Brewer, P.B.; Friml, J. Spatiotemporal asymmetric auxin distribution: a means to coordinate plant development. Cell Mol Life Sci. 2006, 63: 2738–2754.

Materials and Methods.

2.1 Provide more details of the MS media used, product number and company.

Added Duchefa Co. product number M0221.0050

Check on abbreviation for hours and apply to whole of manuscript.

Checked and modified to h

2.2

Line 103.  Detail the reference by name, don’t use the reference number.

Changed to reference No 16

2.4

Line 120-121 appear to say the same thing as Line 132.

Line 132 was deleted

Line 125.  Days should be day

Corrected to day

This section is missing a lot of information.

*We added a complete paragraph and thank the reviewer for pointing our mistake.

  • PCR conditions
  • What is the house keeping gene and primers for the qPCR

The housekeeping gene was 18S ribosomal RNA, the primers were added. To the text

  • pPCR details
  • Are primers designed to only amplify cDNA? i.e. do they cross introns.

The primers are for cDNA that was synthesized from the RNA isolated

  • What method was used for calculating the qPCR results, ΔΔCt?
  • How was the fold change calculated?

The relative abundance of the examined gene transcripts was calculated by the formula: 2(CT_examined gene-CT_reference gene), where CT represents the fractional cycle number at which the fluorescence crosses a fixed threshold.

  • What are the reps, technical or biological? qPCR usually has 3 technical and you then have your biological reps.  Explain how you handled the results.

Three technical replicates were performed for every biological repeat, three biological repeats for each condition. The qRT-PCR analyses were done using the Rotor-Gene Q detection system analyzed with the Rotor-Gene 6000 software (Qiagen Corbett Life Science, Dusseldorf, Germany).

  • Primers for NO synthase genes and NO reductase genes are not detailed. Also explain how the primers discern between the different alleles.

We did not perform qPCR on the NO synthase genes and NO reductase genes.  The data presented are from the Differential Gene Expression experiment.  A clarification was added to the figure legend.

  • Provide more information about where the sequence information came from for design of primers.

Primers were designed based on the tobacco genome in Sol Genomics Network (Sol Genomics Network), gene analysis was performed using the annotation in this database.

  • Were PCR and qPCR products sequenced to confirm correct gene amplified? Espcially for the allele expression results.
  • Except for the allele expression analysis, the PCR products were not sequenced but compared to known controls. The DEG (Differential Gene Expression) results were based on sequence analysis of mRNA so sequencing data existed to compare gene expression.
  •  

*The following paragraphs were added to the texts in Methods section 2.4:

All PCR primers used throughout this study were designed based on the tobacco genome in Sol Genomics Network (Sol Genomics Network) and purchased from Hy Labs Ltd. (Rehovot, Israel). Real-Time PCR (qRT-PCR) analyses were done as described by Schreiber et al. [22].

The PCR reactions were performed in a T-GRADIENT thermal cycler (Biometra, Analytik Jena, and Gottingen, Germany).

Total RNA was extracted from seals and flowers of tobacco using the TRIzol reagent system (Invitrogen Corp, Carlsbad, CA). Genomic DNA contaminants were digested with TURBO DNA-free DNAase (Ambion Inc, Austin, TX, USA). The remaining RNA was then used as the template for cDNA synthesis using the Masterscript cDNA synthesis kit with random hexamer primers (KAPA Biosystems, Woburn, MA, USA).

The qRT-PCR analysis was performed using the KAPA SYBER FAST Master Mix (KAPA Biosystems, Woburn, MA, USA); DNA sequences complementary to tobacco 18S ribosomal RNA were used as a reference.

Three technical replicates were performed for every biological repeat, three biological repeats for each condition. The qRT-PCR analyses were done using the Rotor-Gene Q detection system analyzed with the Rotor-Gene 6000 software (Qiagen Corbett Life Science, Dusseldorf, Germany). The relative abundance of the examined gene transcripts was calculated by the formula: 2(CT_examined gene-CT_reference gene), where CT represents the fractional cycle number at which the fluorescence crosses a fixed threshold.

There are no materials and methods for the Nitric Oxide experiments or the Positional experiments or how choloroplast numbers were coutned.

The following paragraphs were added to the texts in Methods section 2.6:

Chlorophyll content and chloroplast numbers were measured according to Kolotilin et al. [25]. Each time point was the average of five replicas.  The area of leaf segments on various media was measured using ImageJ free software [26]. Each time point is the average of 10 measurements + SE.

 NO staining at the border of leaf segments placed on regeneration medium was done by incubating the leaf segment for 30 min prior to visualization in 50 mM MES buffer pH 5.6 and 1 mM DAF-2DA (4,5-diaminofluorescein-2 diacetate).  Stain intensity per cell was analyzed by using ImageJ free software [26]. The effect of NO on regeneration was done by placing leaf segments on Reg medium containing NO donors (20 mM, S-Nitroso-N-acetyl-DL-penicillamine; 5 mM, Molsidomine), NO scavenger (100 mM, PTIO), and NO synthesis inhibitor (5 mM, Diphenyleneiodonium).

Results

There are some references within the results.  Please confirm with the journal is this is acceptable.  The use of references shows points of discussion that may need to be moved to that section; unless is it confirming experimental protocol.

In places where the references in the results were added as discussion part, they were removed. See line 260.

Line 208.  There is no data to say that chlorophyll content ‘continued to increase.’  Results are only shown for up to 10 days.

The sentence was modified to “starts to increase.”

Line 218 – 222.  This appears to be a summary of the following paragraph, is it needed or should be moved to after the detailed paragraph?

We thought it is needed as a kind of summary to give the punchline at first.

Figure 2.  The images should be resized so that the scale bar is similar for all of them.  This will allow easier comparision between images.  i.e. cannot really compare day 0  to day 5 as the size of the day 0 image is very small.

Line 227.  How did you measure the chloropolast numbers?  (Details should be in Materials and Methods)

A reference was added to the Methods.

Line 242, Figure 3 should be moved to the end of the sentence.

In the copy I got, Figure 3 was missing, so I added it again in the indicated place.

Line 243.  What type of DNA?  gDNA would confirm the integration of the transformation event, cDNA would confirm transcription.

We tested for integration, so genomic DNA was used, the word genomic was added before DNA.

Figure 3.  The bands on the gell are very hard to see.  It appears that cDNA of the transgene was detected in the red leaf segements of plants I, H, G and E only, not F.  And in the green leaf segments for plants G and E only, not I, H and F.  Would it be possible to change the image to white and black instead of black and white, making the bands black.  Thank you for including the positive and negative controls.

We tested for integration, and sometimes, the bands are weak.  Lanes F and G have a weak band.  Leaf G had a very small red zone that might explain why there us a weak band (very little DNA)

Line 253.  Reword sentence to remove We.

The word We was removed

Figure 4. There is no reference to 4E in the results, suggest removing it.  If incorporated the pigmented leaves may need to be highlighted with an arrow.

We added arrows as the reviewer suggested.  And added to the legend “Arrows point to the colored leaves in the plate.”

3.4

Line 279 and 282.  Figure 5C

We amended it to 5C instead of 5.

Line 285.  The transcript levels increased and decreased, not just increased.

We amended the sentence to “The transcript level of Nitrate reductase 1-3 increase and then decreased in the induction period without any difference between the various alleles (Figure 5E).”

Figure 5.

Detail what the error bars are, standard error or standard deviation?  Include number of samples to calculate averages and error bars.

Error bars are used, and the following sentence was added to the figure legened “Error bars indicated standard error of 50 cells in B; 20 segment in 3 replicated in C, and 3 biological replicates in D and E.”

5B.  The mean intensity of what?  How is this measured?  What happened in Day 5?  The day 5 results significantly alter the inferred lines.  If day 5 was not included the inferred lines would look different. 

The intensity of each cell was measured by ImageJ program, as stated in Methods.  Day five was high for the DiPhenyl treatment, and it did change the line somewhat. Still, the conclusion that Diphenyl inhibits NO production correlated with inhibition of regeneration does not change if we omit the day 5 point.

5C. What is the P level for the comparison?

p(f) was 7.65x10-35 using the Tukey test.  The P-value was added to the figure legend.

Figure 6B.  Correct spelling of y-axis.

The y-axis was corrected. The words fraction and connections are spelled correctly.

Figure 6. Detail what the error bars are, standard error or standard deviation?  Include number of samples to calculate averages and error bars. 

The following sentence was added to the legend “Error bars indicated standard error of 50 cells in B; and 3 biological replicates in C.”

Figure 7.  I appreciate that this is a sample of the transcript analysis, but the processes listed mean nothing as you don’t know what bar they represent.  Could you focus on the top 20 processes or the 10 highest/lowest processes and show their bars and include the full information in a supplementary figure/table.  This figure only shows that there was differences in expression profiles.

As suggested by the reviewer, we show a sample of 20 top processes on day 4, and the full processes for all the days are in supplementary data, figures S3 to S5.

Line 338-342.  Data not shown for results presented.  Include in a supplementary table.

Data is presented in the supplementary section, as suggested by the reviewer.

Line 343-348  Data not shown for results presented.  Include in a supplementary table.

Data is presented in the supplementary section, as suggested by the reviewer.

Figure 8.  Fix figure, Day 4 and day 7 colours?  Where are the positive and negative controls for the PCR gel.

The colors were changed.  As for positive control, we did not have a plasmid for positive control; this is why we used a cycle calibration curve with predicted size based on the sequence.  

3.6

Need information about how plates were prepared.  Was the leaf segment placed on the agar and then turned upside down, or placed under the agar?  What was the lgiht conditions?  Could light be a factor, not gravity?

Segments were placed on the agar, and the plates were turned upside down. Light conditions were identified as the agar was transparent. The plates were very well illuminated, and the experiment was repeated in several growth rooms with slightly different lighting conditions and always gave the same result.

Figure 10.  Provide further detail as to how % shooting was calculated.  Maybe include values in a supplementary table.

The following stamen was added to the legend in figure 10, “percent shooting was calculated as the number of segments that produced at least one shoot out of a total number of the segment in a plate. Results are the mean + se of five repetitions for each treatment.”

Line 391. > 10 days.

It was corrected as suggested.

Line 400-401.  Do you mean that after 4 days exposure the response is irreversible?

Yes, after four days, the tissue is committed.

Lin 410. Different reference syle.

We could not find to the reviewer meant by this remark

Discussion.

The discussion is more a summary of the results and does not show much comparison with published literature.  Published literature may be scarce. 

The discussion is extended and the order of things changed.

Why did you do the nitric oxide and positional experiments?  Was there other literature that suggested you should test these? 

Another cellular messenger molecule is NO (Nitric oxide) that regulates growth processes such as vegetative and generative development, seed germination, root growth, gravitropism, flowering, and fruit ripening [18]. Furthermore, NO also participates in the abiotic stress responses of plants [18]. The growth regulating effect of NO is caused by the auxin-NO interplay and cytokinin interaction, regulating cell division [19] and shoot regeneration [20].  We tested the effect of NO modifiers on shoot regeneration to learn about the possible role of NO as a messenger in shoot regeneration. We show here that NO has a role in shoot regeneration and that inhibiting NO production inhibits shoot regeneration (Fig. 5). All the information that the leaf cells receive but starting to communicate with their neighbours is then somehow translated to induction of shoot regeneration in the competent tissue.

How does the change in gene expression compare to other plant expression profiles? 

The point addressed in discussed “Global gene expression of particular genes of interest analyses could not pin down a single or specific process necessary to induce shoot regeneration (Figs. 7, 8, 9, S3 to S9). This observation that a multitude of genes change when global gene expression is studied during regeneration was shown in Arabidopsis as well [8].  It seems that shoot regeneration is a complex phenomenon that involves many processes and is not triggered by a single gene.”

Any ideas how to ensure regenerated plants originate from single transformation events?

It seems that the way to overcome the chimerism problem is using at least T1 seeds and in targets that do not produce seeds, be aware that chimerism can obscure the results

No discussion on the differences in expression of gene families.  A general statement with examples and correct references of how complex gene families are and how expression can be dependent on tissue/cell localisation, plant development stage, environmental conditions to name a few.

All the gene families check in this study show a similar pattern. Several gene family members increase during the induction period and several decreases, and most do not change. It seems that shoot regeneration is a complex phenomenon that involves many processes and is not triggered by a single gene.

No discussion on the lack of specific biological processes being identified.  I would not expect a single process as within the clump of cells regenerating into a plantlet there are a multitude of cell types being formed. Unlike a matuer plant where leaf cells are clearly defined from stem cells.  There could be further discussion about this.

There was no single process within the cells regenerating into a shoot at the induction period.  One has to remember that the period checked is prior to leaf or meristem formation, and no obvious visual difference can be ascertained.  Once a protomeristem is form we can define localization of certain processes and genes.  At the stages we checked the cells looked alike.

Round 2

Reviewer 1 Report

1). Manuscript ID: Plants-1037967

2). Manuscript Title: Shoot regeneration is not a single cell event

3). Comments: Thank you for modifying the manuscript according to our recommendations.

Reviewer 2 Report

No Further comments.  Thank you for addressing my points.